# Hypoxic Cardioprotection by New Antihypertensive Compounds in High Salt-Diet Hypertensive Rats: Glucose Transport Participation and Its Possible Pathway

**DOI:** 10.3390/ijms25168812

**Published:** 2024-08-13

**Authors:** Manuel A. Hernández-Serda, Aldo Y. Alarcón-López, Víctor H. Vázquez-Valadez, Paola Briseño-Lugo, Pablo A. Martínez-Soriano, Viridiana Leguízamo, Nalleli Torres, Rodrigo González-Terán, Luis A. Cárdenas-Granados, Fausto Sánchez Muñoz, Emma Rodríguez, Claudia Lerma, Alejandra María Zúñiga Muñoz, Enrique Ángeles, Roxana Carbó

**Affiliations:** 1Departamento de Ciencias Químicas FES Cuautitlán, UNAM, Av. 1° de Mayo S/N, Santa María las Torres, Campo Uno, Cuautitlán Izcalli 54740, Estado de México, Mexico; serda@cuautitlan.unam.mx (M.A.H.-S.); yoshalar@gmail.com (A.Y.A.-L.); 2Departamento de Ciencias Biológicas, FES Cuautitlán, UNAM, Av. 1° de Mayo S/N, Santa María las Torres, Campo Uno, Cuautitlán Izcalli 54740, Estado de México, Mexico; hugounam83@gmail.com (V.H.V.-V.); paolaedithb@yahoo.com.mx (P.B.-L.); viridiana.leg.glez@gmail.com (V.L.); nallelitorresgarduno21@gmail.com (N.T.); rodrigo_g_teran@hotmail.com (R.G.-T.); 3QSAR Analytics SA de CV. Tempano 10, Colonia Atlanta, Cuautitlán Izcalli 54740, Estado de México, Mexico; 4Laboratorio de Química Medicinal y Teórica, Departamento de Ciencias Químicas, FESC, UNAM, Av. 1° de Mayo, Col. Sta. María las Torres, Cuautitlán Izcalli 54740, Estado de México, Mexico; parturomart@cuautitlan.unam.mx (P.A.M.-S.); aalfonsocardenas@gmail.com (L.A.C.-G.); angeles@unam.mx (E.Á.); 5Departamento de Inmunología, Instituto Nacional de Cardiología Ignacio Chávez, Juan Badiano #1, Col. Sección XVI, Tlalpan, Ciudad de México 14080, Mexico; fausto22@yahoo.com; 6Laboratorio de Medicina Traslacional UNAM-INC, Instituto Nacional de Cardiología Ignacio Chávez, Juan Badiano #1, Col. Sección XVI, Tlalpan, Ciudad de México 14080, Mexico; emarod2@yahoo.com.mx; 7Departamento de Biología Molecular, Instituto Nacional de Cardiología Ignacio Chávez, Juan Badiano #1, Col. Sección XVI, Tlalpan, Ciudad de México 14080, Mexico; dr.claudialerma@gmail.com; 8Departamento de Biomedicina Cardiovascular, Instituto Nacional de Cardiología Ignacio Chávez, Juan Badiano #1, Col. Sección XVI, Tlalpan, Ciudad de México 14080, Mexico; mvzalemar@yahoo.com.mx

**Keywords:** cardiomyocytes, high-salt hypertension, changrolin, glucose transporters, AMPK, new drugs

## Abstract

Hypertension (HP) is a health condition that overloads the heart and increases the risk of heart attack and stroke. In an infarction, the lack of oxygen causes an exclusive use of glycolysis, which becomes a crucial source of ATP for the heart with a higher glucose uptake mediated by glucose transporters (GLUTs). Due to the unpleasant effects of antihypertensives, new drugs need to be researched to treat this disease. This study aimed to evaluate the cardioprotective effect of three novel antihypertensive compounds (LQMs, “Laboratorio de Química Medicinal”) synthesized from Changrolin under hypoxic conditions with the participation of two primary cardiac GLUT1 and GLUT4 using a high-salt diet HP model. The model used a diet with 10% salt to increase arterial blood pressure in Wistar rats. In isolated cardiomyocytes from these rats, glucose uptake was measured during hypoxia, evaluating the participation of GLUTs with or without the animals’ previous treatment with LQM312, 319, and 345 compounds. In silico calculations were performed to understand the affinity of the compounds for the trafficking of GLUTs. Results: Control cells do shift to glucose uptake exclusively in hypoxia (from 1.84 ± 0.09 µg/g/h to 2.67 ± 0.1 µg/g/h). Meanwhile, HP does not change its glucose uptake (from 2.38 ± 0.24 µg/g/h to 2.33 ± 0.26 µg/g/h), which is associated with cardiomyocyte damage. The new compounds lowered the systolic blood pressure (from 149 to 120 mmHg), but only LQM312 and LQM319 improved the metabolic state of hypoxic cardiomyocytes mediated by GLUT1 and GLUT4. In silico studies suggested that Captopril and LQM312 may mimic the interaction with the AMPK γ-subunit. Therefore, these compounds could activate AMPK, promoting the GLUT4 trafficking signaling pathway. These compounds are proposed to be cardioprotective during hypoxia under HP.

## 1. Introduction

An estimated 24% of the adult population suffers from HP, a chronic medical condition. A person with blood pressure greater than 140/90 mmHg is considered hypertensive. The presence of comorbidities, such as obesity or insulin resistance, as well as unhealthy habits, such as high alcohol intake, high salt intake, and stress, are considered predisposing factors to generate HP. Lifestyle changes can improve blood pressure, although drug treatment is often necessary when such changes are not effective. Many patients diagnosed with HP take one or a combination of various antihypertensive medications, and the effectiveness of therapeutic effects depends on the predisposing factors mentioned above [1].

Some drugs are used to prevent HP, including beta-blockers, angiotensin-converting enzyme (ACE) inhibitors, thiazide diuretics, angiotensin receptor blockers, and calcium channel blockers [2]. Nevertheless, they all have disadvantages, such as hyperkalemia, fatigue, coughing, or dizziness [3]. If left untreated, HP can cause cardiac pressure overload, hypertensive heart disease, hypertrophy, and remodeling of the cardiac cells and the vasculature [4].

Heart artery obstruction due to remodeling or lipid obstruction leads to an infarction, creating a hypoxic environment for cardiomyocytes. Under resting conditions, heart metabolism is approximately 70% dependent on lipid oxidation, and the remaining energy is produced by glucose oxidation. Meanwhile, during hypoxia and excessive muscle contraction, heart cells increase glycolysis to fuel their metabolism, which has been associated with better heart survival [5].

The cell glucose transport mechanism has been studied for decades since 1939, with Einar Lundsgaard’s observations and the formulation of the translocation hypothesis in 1980 [6,7]. Glucose is internalized into cells principally [8] via facilitative glucose transporters called GLUTs, which constitute a family of 14 proteins, and by Na^+^/coupled glucose transporters (SGLTs), which belong to the family of membrane proteins encoded by SLC2 genes [9]. The heart uses GLUT1, GLUT4, and SGLT1 [10].

GLUT1 is widely expressed as the primary mediator of glucose uptake in all tissues, and GLUT4 is present primarily in insulin-dependent tissues, such as muscle (skeletal and cardiac) and adipose tissues [6]. The insulin-dependent GLUT4 trafficking pathway has been extensively studied [9,11]. Glucose uptake within cardiomyocytes occurs mainly through glucose transporters through an insulin-dependent or insulin-independent process. Insulin-stimulated and contraction-dependent glucose uptake occurs via GLUT4, while insulin-independent glucose uptake occurs primarily through GLUT1 [12,13]. In addition, other pathways that can modify glycemic control have also been investigated, including the path-regulating exercise-stimulated glucose uptake [14], the effects of angiotensin II as a regulator [15], transitory hyperglycemia [16], and the hypoxia-induced pathway [17]. Although these stimuli also cause GLUT1 trafficking, GLUT4-mediated uptake is likely the most important way the heart takes in more glucose in these situations [18].

The insulin-dependent mechanism acts through the phosphatidyl-inositol-3 kinase/protein kinase B or Akt2 (PI3K/Akt2) pathway, incorporating GLUT4 into the plasma membrane [15]. However, when energy demand is high, such as during exercise or hypoxia, transport occurs through the insulin-independent AMP-activated protein kinase (AMPK) pathway [19].

AMPK is a molecule particularly sensitive to deregulated adenosine monophosphate AMP/ATP ratios and senses an increase in cellular AMP levels due to cellular ATP levels depletion. The glycolysis process is increased to maintain intracellular ATP concentrations. It becomes an essential source of ATP during ischemia. The main action of AMPK is to phosphorylate targets that turn off nonessential anabolic processes that consume ATP and turn on catabolic pathways that produce ATP, compensating for ATP deficits [20]. AMPK is a crucial regulator of glucose uptake in cardiac myocytes [21]. Previous studies support that the AMPK-mediated inhibition of protein synthesis and cellular energy preservation can have an antihypertrophic effect [13].

All these mechanisms, culminating in the translocation of GLUT4 to the membrane, involve numerous proteins that integrate two intricate systems: signal transduction and vesicular transport [7]. Some intracellular vesicles (GSVs) store specific glucose transporters, such as GLUT4 and GLUT1 [20,22].

Research efforts have been directed toward the development of drugs that target these proteins, willing to achieve beneficial pharmacological effects in a variety of conditions, including type 2 diabetes mellitus [23], cardioprotective effects [24,25], and potential cancer therapies [26].

Changrolin (2,6-bis[pyrrolidin-1-ylmethyl]-4-[quinazolin-4-ylamino] phenol) is an antimalarial and antiarrhythmic drug derived from *Dichroic febrifuga*, an essential herb in traditional Chinese medicine [27,28]. In the “Laboratorio de Química Medicinal” (LQM) of the “Drug Design in Medicinal Chemistry Program” of the “Universidad Nacional Autónoma de México” (UNAM), the Changrolin molecule was modified by taking the phenol and methyl pyrrolidine rings as structural requirements to show cardiovascular effects and changing the pyrrolidine rings to methylthiomorpholine rings. In our experience, methylmorpholine phenol and methylpiperidin phenol derivatives show cardiovascular effects [29], and thiomorpholin phenols have antimycobacterial properties [30,31]. Our group also reported new methylthiomorpholine phenol compounds with cardiovascular effects, considering that new antihypertensive drugs are necessary to reduce adverse side effects [32].

Our experimental studies show that LQMs from the 300 series exhibit affinity for the same activity site in the angiotensin-converting enzyme (ACE), which converts angiotensin I into angiotensin II. Angiotensin II is a vasoconstrictive peptide that plays a role in high arterial blood pressure, heart failure, and myocardial infarction. Captopril was selected due to its known characteristics and because it is a drug that explicitly blocks ACE, and these compounds share the same affinity for ACE [33].

In silico studies were conducted to study, at the molecular level, AMP-activated protein kinase (AMPK) as a possible mediator for these compounds with cardioprotective activity by promoting glucose transporter mobilization.

This study aims to evaluate the cardioprotection of these compounds mediated by the heart-specific glucose transporters during a metabolic change present in a hypoxic event, besides their possible protection due to their antihypertensive action.

## 2. Results

### 2.1. HP Model

To perform a hypertensive model, 250 g rats ate a high-salt diet (10%) for four weeks. The HP animals showed high systolic blood pressure and higher food consumption due to the addictive characteristic of salt. Therefore, they drank much more water and urinated more. The data of HP model are shown in Table 1. To confirm that the high blood pressure in the animals was due to the high-salt diet, they were deprived of it for another four weeks. HP animals developed high arterial blood pressure and kept it even if they stopped consuming a high-salt diet.

### 2.2. Antihypertensive Effect of Captopril and LQMs

As a first approach to the compounds, they were tested as antihypertensives. A classic ACE inhibitor (Captopril) was used as a control, and the new compounds lowered blood pressure, with Captopril being the one with the most significant antihypertensive effect. The treatment with LQM-312 showed a slightly higher blood pressure in the control animals but did not have biological significance (Figure 1).

### 2.3. Cardiomyocyte Glucose Uptake

Isolated cardiac cells were used to explore the metabolic cardioprotection of the compounds and the participation of the glucose transporters. Glucose uptake shows the cells’ behavior when they confront different stressors and hypoxic conditions, which were used to recreate some conditions, like a heart attack. After the extraction procedure, cell viability was measured, and 67% of live cells were obtained in each experiment. There was no significant difference due to the treatment, even during hypoxia. The cells showed typical striations characteristic of cardiac cells and tolerated physiological calcium concentrations [34,35]. The participation of GLUTs in glucose uptake was determined by incubating cells with specific antibodies raised against the cytoplasmatic region of the glucose transporters, preventing glucose from entering and indirectly evaluating the transporter participation. In previous studies [36,37], a positive control was used by incubating the cells with a selective glucose transporter blocker (cytochalasin B (0.1 µM/L; Sigma, St Louis, MO, USA)) [38], and a negative control was used by incubating the cells with a goat serum diluted in Tyrode at the same concentration as the specific antibodies used. Those experiments were performed to assess the specificity of the antibodies and the intraexperiment reliability. 

#### 2.3.1. Controls

Under oxygenated control conditions, rat cardiomyocytes showed a basal Tyrode glucose uptake of 1.84 ± 0.09 mM of glucose per 4 × 10^4^ cells/hour (approximately 100 mg of tissue (mM/100 mg/h)), comparable to previous studies [36,37].

During hypoxia, the cells changed their metabolism mainly to the glucose-dependent pathway (2.67 ± 0.1 mM/100 mg/h).

When the specific antibody against the transporters was applied, glucose consumption decreased since the antibody prevented glucose from entering, which means indirectly that the transporter is responsible. On the other hand, when consumption was not affected despite the presence of the antibody, this transporter did not participate in consumption.

During oxygenation, glucose uptake was mediated only by GLUT1, and under hypoxia, both transporters participated (Figure 2, panel A black bars).

#### 2.3.2. Hypertensives

During oxygenated Tyrode conditions, hypertensive animal cells tended to have a higher uptake than the controls (2.38 ± 0.24 mM/100 mg/h).

Still, they did not respond during hypoxia (2.33 ± 0.26 mM/100 mg/h) by changing their metabolism to a protective glucose-dependent pathway (Figure 2, panel B). Hypertensive cells did their uptake with the participation of both transporters during oxygenation and hypoxia (Figure 2, panel B black bars).

### 2.4. Captopril Effect on Glucose Uptake

To assess the effect of the well-known antihypertensive Captopril on heart metabolism, the cells of pretreated animals were allowed to consume glucose and the transporters involved were evaluated.

#### 2.4.1. Controls

When the control animals were treated with Captopril, their cardiomyocytes consumed less glucose under oxygenated conditions (1.45 ± 0.15 mM/100 mg/h) than under Tyrode conditions, but not significantly.

Although Captopril animals’ cardiomyocytes consumed significantly less glucose in hypoxia than hypoxic Tyrode cells, the cells tended to consume more glucose in hypoxic conditions, which means they could make the saving metabolic transition as Tyrode cells do under the same circumstances. Under control conditions, Captopril allowed the mobilization of GLUT4 during oxygenation and hypoxia (Figure 2, panel A blue bars).

#### 2.4.2. Hypertensives

Our study found that Captopril pretreatment in hypertensive oxygenated cells tended to consume less glucose than hypertensive Tyrode cells, but this difference was not statistically significant.

During the hypoxic event, the metabolic response was significantly recovered in cells with the Captopril treatment compared to its oxygenated Captopril-treated cells. This effect significantly differs in the glucose uptake with the Captopril treatment compared to Tyrode cells in hypoxia.

On the other hand, in HP, both transporters participated in glucose uptake in both conditions, as in the control cells (Figure 2, panel B, blue bars).

Hypertensive cells from Captopril animals uptake more glucose than those with Captopril in the controls, oxygenation, and hypoxia (Figure 2, panels A and B blue bars).

**Figure 2 ijms-25-08812-f002:**
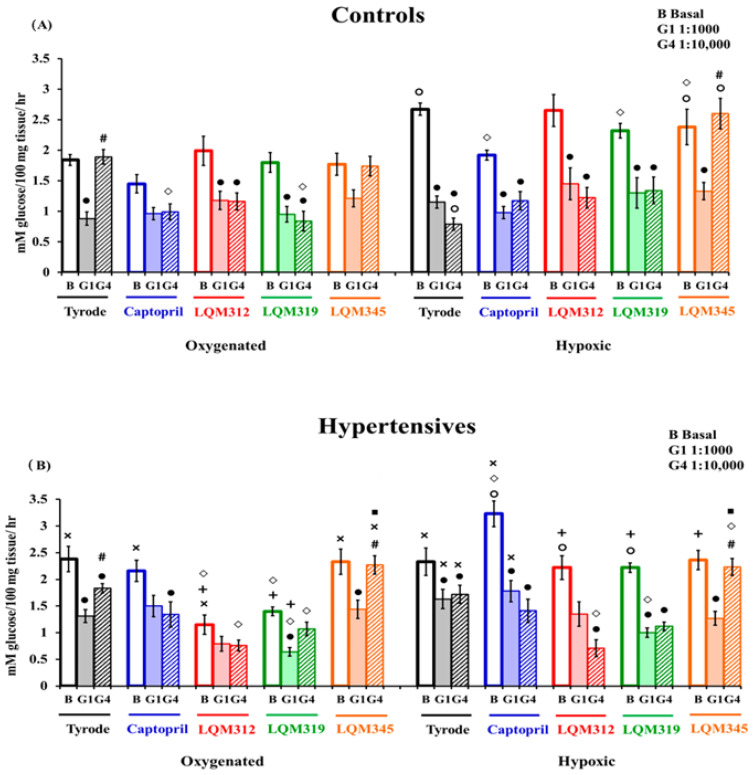
Glucose uptake of the control and hypertensive groups with and without hypoxia via GLUT1 and GLUT4. (**A**) Control cell uptake under the abovementioned conditions. (**B**) Hypertensive cell uptake under the abovementioned conditions. The values are expressed as mean ± SEM, n = 6, except for the Tyrode groups, n = 24. Four-way ANOVA (factors are group, treatment, hypoxia, and transporter) with post hoc comparisons corrected by the Bonferroni method: **◦**: *p* < 0.05, statistically significant vs. its corresponding oxygenated group; •: *p* < 0.05, statistically significant antibody effect vs. its basal uptake in both conditions separately; **#**: *p* < 0.05, statistically differences between GLUT 1 and GLUT 4 antibodies in all cases; **◇**: *p* < 0.05, a statistically significant effect of the compounds vs. its corresponding Tyrode group in both conditions; **×**: *p* < 0.05, statistically significant controls vs. hypertensives in their corresponding groups; **+**: *p* < 0.05, statistically significant compounds vs. Captopril in their corresponding group in both conditions; ■: *p* < 0.05, statistically significant vs. all GLUT4 groups in both conditions separately. G1: 1000 is GLUT1 antibody dilution and G4: 10,000 is GLUT4 antibody dilution.

### 2.5. The LQM312 Effect on Glucose Uptake

To determine the LQM312 effect on cardiac cell metabolism, glucose uptakes in the cells of the pretreated animals and the transporters involved were evaluated.

#### 2.5.1. Controls

When the control animals were treated with LQM312 cardiomyocytes, glucose uptake on oxygenation was almost equal to that of the control cells without pretreatment (1.9 ± 0.24 mM/100 mg/h) and showed a tendency to uptake more glucose under hypoxic conditions, but it didn’t come out significant. As with Captopril, LQM312 showed that the two GLUTs participate in oxygenation and hypoxia (Figure 2, panel A red bars).

#### 2.5.2. Hypertensives

Our findings show that, during HP, the cells from animals treated with LQM312 take up much less glucose in oxygenation than the control cells and less than their hypertensive Tyrode experiments.

However, LQM312 promotes metabolic recovery during hypoxia in HP, but it is not as significant as Captopril. LQM312 uses both transporters to uptake glucose, but GLUT4 participates more strongly in hypoxic hypertensive cells (Figure 2, panel B, red bars).

### 2.6. The LQM319 Effect on Glucose Uptake

To investigate the effect of LQM319 on heart metabolism, the cardiac cells of pretreated animals were stimulated to consume glucose and the transporters involved were evaluated.

#### 2.6.1. Controls

LQM319 has the same effect on glucose uptake in oxygenation (1.8 ± 0.16 µg/g/h) as the Tyrode group, although the tendency to consume more glucose in hypoxia is not significantly different. This compound’s impact is not different from that of Captopril and LQM312, resulting in both transporters’ participation in oxygenation and hypoxia (Figure 2, panel A green bars).

#### 2.6.2. Hypertensives

In HP pretreated with LQM319, cells consume significantly less than their hypertensive Tyrode cells. The two transporters are responsible for glucose consumption in this condition. However, GLUT1 participates more significantly during oxygenation.

During hypoxia, this compound’s protective effect allows the two transporters to increase glucose uptake, but not as efficiently as Captopril (Figure 2, panel B green bars).

### 2.7. The LQM345 Effect on Glucose Uptake

To ascertain the LQM345 effect on glucose metabolism of cardiomyocytes from pretreated rats, isolated cells were put to consume glucose, and their transporters were evaluated.

#### 2.7.1. Controls

During LQM345 pretreatment, cells consume a basal amount of glucose similar (1.7 ± 0.18 mM/100 mg/h) to that of the Tyrode, Captopril, LQM312, and LQM319 during oxygenation.

During hypoxia, cells can change their metabolism to glycolytic. However, as this does not allow GLUT4 translocation to the membrane, only GLUT1 handles the glucose entrance (Figure 2, panel A orange bars).

#### 2.7.2. Hypertensives

Pretreatment with LQM345 does not affect glucose uptake in oxygenation or hypoxia, and GLUT4 participation was absent in any experimental treatment during HP (Figure 2, panel B orange bars).

Given the concentration of these antibodies (1:10,000), the antibodies did not affect hypertensive cells during hypoxia, which prompted new experiments in which the concentration was increased 20-fold (1:500). However, there was no effect on GLUT4 (Figure 3).

This compound has a very different effect on glucose uptake than the other tree compounds tested.

### 2.8. Computational Studies of Molecular Interactions

Given the results of the glucose transporters’ participation in the action of these new compounds, the concern arose to elucidate the mechanism by which this phenomenon could occur. In our experimental conditions, glucose transport occurs through the insulin-independent AMP-activated protein kinase (AMPK) pathway [19]. AMPK is particularly sensitive to cellular ATP levels, and the glycolysis process is increased to maintain intracellular ATP concentrations, becoming an essential source of ATP during ischemia. AMPK was chosen for its crucial role in promoting GLUT translocation to the membrane, glucose uptake, and glycolysis [39].

An in silico study was carried out on the possible interactions of the compounds with a molecule that participates in the movement of transporters to the membrane.

A three-dimensional model of the αβγ heterotrimer protein AMPK in *Rattus norvegicus* was generated. Figure 4 illustrates the generated three-dimensional model along with its corresponding linear representations. The figure also highlights the specific locations where molecular docking was conducted.

The entire AMPK protein was modeled, with its three well-identified interaction sites: the ATP-site inhibition pocket, the AMP-site activation pocket, and the allosteric activation site. Blind docking was performed to assess the energy differences among the three LQM compounds under investigation at these sites.

Based on the docking results, the estimated binding affinities of the three LQM molecules for each site on the AMPK protein were calculated. However, as it was found that none of the compounds had binding modes at the allosteric activation site, we only present results for the AMP and ATP sites in Table 2.

Captopril binds to the AMPK γ-subunit. Similarly, LQM312 also binds to the γ-subunit and shows a certain, albeit lower, affinity for the ATP site between the α-subunit and β-subunit. The compound LQM319 did not show interaction modes at the activation site of the AMPK γ-subunit but showed a strong affinity for the inhibitor site in the α-subunit. This finding suggests that this compound’s cardioprotective effect does not occur through the activation of AMPK at the AMP γ site. In the case of the ligand LQM345, most of the binding modes were situated at the ATP site of the α-subunit, indicating a high probability that this ligand will bind to the inhibition pocket site.

Two molecular dynamics simulations, one with Captopril and another with LQM312, were performed to figure out whether the interaction defined by the docking would persist over time and under the physiological conditions of the cell environment. It was confirmed that the ligand remained at the docking site.

Once the system reached equilibrium, the parameters of root mean square deviation (RMSD) and root mean square fluctuation (RMSF) were calculated, as presented in Figure 5. The RMSD graph allows us to analyze changes in the protein’s conformation and folding relative to its initial position. The variation in the RMSD is similar in both simulations. The RMSF graph assesses amino acid flexibility during the simulation, reflecting the fluctuation of atomic positions concerning the simulation’s overall average.

Figure 6 shows the interactions that stabilize the γAMPK-Captopril and γAMPK-LQM312 complexes. Both molecules have hydrogen-bonding acceptor interactions with methionine 84.

ADMET property predictions were performed using the SwissADME [40] server to evaluate the Lipinski rules and estimate the LQM compounds’ permeability properties. The results are summarized in Table 3.

Considering the characteristics of the molecules used and the number of drugs that exhibit good absorption and permeation properties, it is evident that most of these molecules adhere to Lipinski’s “Rule of Five”. Furthermore, the molecules LQM 312, LQM 319, and LQM 345 exhibit properties that could result in a high probability of absorption and permeation. These parameters are crucial when considering the site of action and the necessary membrane crossing for their effect.

## 3. Discussion

Our findings show the effects of Captopril and the LQM compounds on blood pressure. As anticipated, we observed a reduction in blood pressure using the high-salt diet HP model. Furthermore, we evaluated the cardioprotective effect of the new compounds by assessing their performance during hypoxia, focusing on the participation of the heart’s GLUTs.

Cardiac-isolated myocytes were used to observe the cardioprotective glucose uptake in hypoxia with the participation of the glucose transporters. Regarding the behavior of glucose consumption of the cells extracted from the control and hypertensive animals, cardiac cells from the control animals under basal conditions showed glucose uptake as expected [41]. The cells made the metabolic shift during hypoxia with the activation of GLUT4.

Meanwhile, the cells from the hypertensive animals exhibited increased glucose consumption during oxygenated conditions [42], which was promoted by heart adaptation to extra work compared to the Controls. And they were unable to undergo the expected metabolic shift during hypoxia. An activation of GLUT4 was observed during oxygenation, which may indicate the onset of hypoxia in these cells [43], the product of energy fatigue produced by the pressure overload of the heart in hypertension [4]. Previously, it was reported that HP causes glucose intolerance and increases in GLUT4 [43] and is regulated by different tissue- and stimulus-specific mechanisms [44]. Another possible explanation for the unresponsiveness of hypertensive cells could be that, in hypoxia, cells tend to accumulate Na+ [45], and our HP model was formed by a high salt uptake.

Captopril, a selective angiotensin-converting enzyme (ACE) inhibitor with a well-established role in clinical practice, was chosen as the positive control in our study. It prevents chronic heart failure, reduces systolic ventricular demand, and benefits hemodynamics, hypertrophy, and infarct size [46,47], promoting less cardiac effort and lower glucose requirement. Our results show that cells from the control animals pretreated with Captopril tend to consume less glucose and showed the participation of GLUT4 in oxygenation, confirming previous reports where Captopril stimulates skeletal muscle gene expression and synthesis of GLUT4 [43]. During hypoxia in the controls, Captopril acts as oxygenation.

In HP, our results show that Captopril has a cardioprotective effect, maintaining the metabolic stability of the heart in oxygenation and protecting it during oxygen deprivation with a prevalence of GLUT4.

In the control cells with oxygenation, the compounds LQM312 and LQM319 do not show changes and behave like cells from the untreated animals, except for the presence and activation of GLUT4, as Captopril does. Based on these findings and previous reports [48], we propose that these compounds may also work through an affinity to ACE because they behave like Captopril, while LQM345 does not seem to react. On the other hand, our findings show a tendency to improve hypoxic conditions when the control animals are treated with LQM compounds. Only LQM345 allows the metabolic change; however, GLUT4 does not participate as expected.

Another interesting finding is that, when HP is present, during oxygenation, the cells treated with LQM312 and LQM319 decrease their glucose significantly, meaning that they require less glucose as if they were healthy with the participation of GLUT4; however, LQM319 has a significant contribution from GLUT1. LQM345, again, does not seem to have any effect. During hypoxia, both LQM312 and LQM319 shift their metabolism to be more glycolytic using the GLUTs. They reverse the metabolic unresponsiveness of hypoxic cells during high blood pressure, having a cardioprotective effect on hypoxia.

Visualizing all these results, the concern arose to elucidate the mechanism by which GLUT4 is mobilized with these compounds. The possible molecules involved were analyzed, and it was determined that AMPK is a molecule in which the stimuli (cardiac work, hypoxia) to which cardiac cells were exposed converge and is a candidate due to its importance in the glucose metabolism [39].

AMP-activated protein kinase (AMPK) is a serine/threonine protein kinase activated by reducing cellular ATP levels. AMPK activation increases energy-generating pathways and reduces energy-consuming pathways to restore energy homeostasis. Its activation or inhibition specifically influences two members of the Rab GTPase-activating protein (GAP) family, TBC1D1 and the 160 kDa Akt substrate (AS160; also named TBC1D4), proteins associated with membranes containing GLUT that are responsible for glucose uptake changes [11]. AS160 is also subject to AMPK phosphorylation. The AMPK-mediated phosphorylation of AS160 and the translocation of GLUT4 are also involved in contraction- or exercise-induced glucose uptake [49]. However, the AMP/ATP ratio increases under hypoxia-induced stress, activating an alternative pathway, insulin-independent AMPK signaling [14]. Curiously, natural AMPK activation during myocardial hypoxia does not reach its maximum, but the addition of an activator triggers the protein kinase. Acute pharmacological intervention promotes AMPK activation in ischemia, enhancing glucose uptake. It also preserves mitochondrial function, reducing reactive oxygen species (ROS). Therefore, its activation improves post-ischemic heart recovery [50].

We explored the potential binding sites and preferred conformations of the Captopril molecule and the three LQM300 compounds with AMPK through molecular docking techniques. These simulations yield valuable insights into the intermolecular forces governing the interactions and the stable conformations of the resulting complexes. In addition to molecular docking, we conducted molecular dynamics simulations to examine the stability and dynamics of protein-AMPK complexes over time. These simulations allow us to explore how interactions evolve in response to changes in the molecular environment and how associations are upheld at the structural level.

The findings derived from these computational studies will offer essential insights into the nature of interactions involving AMPK and modulator molecules.

Considering that Captopril and LQM312 can replicate the interaction between AMP and the AMPK subunit, as proposed in these in silico studies, they may also enhance AMPK phosphorylation and may activate the GLUT4 trafficking signaling pathway since it has been reported that AMPK activation is responsible for glucose uptake in hearts subjected to ischemia by increasing the intracellular translocation of GLUTs to the membrane through a signaling pathway distinctly different from insulin. This trafficking could be considered a compensatory mechanism to increase glucose uptake rather than protein levels [12], underscoring this kinase’s importance in the heart’s pathophysiology [51].

LQM319 does not have an affinity for AMPK as it does for LQM312 and Captopril, suggesting that it uses another pathway to promote the GLUTs’ recruitment to the membrane.

LQM345 is a piperidinylmethyl phenol with biological activity as an antihypertensive [52], but the present results show that it does not seem to function through ACE [53] or AMPK. Other glucose transporters, such as GLUT11 or GLUT12, which are present in the heart, might be responsible for glucose uptake [54]. SGLT participation is questionable in that SGLT2 is not highly expressed in the heart and is not directly linked to energy production but rather natriuresis and processes such as the production of reactive oxygen species, inflammation, and ion homeostasis rather than glucose transport, as demonstrated through inhibitors of this transporter [55]. SGLT1 is very high in the small intestine and the heart and is also an important glucose transporter with preferential localization in the sarcolemma; however, SGLT1 mediates the increased cardiac glucose uptake in response to insulin and leptin [56].

## 4. Materials and Methods

### 4.1. Chemical Compounds and General Synthesis Methodology

The LQM compounds were prepared from phenol derivatives (1 eq.), morpholine, thiomorpholine, or piperidine (2.1 eq.), and formaldehyde (37%) (2 eq.). They were mixed in a round flask fitted with a condenser, and the mixture was irradiated with infrared light using a medicinal infrared lamp (250 W) under solvent-free conditions [57]. The reaction was monitored by thin-layer chromatography (TLC) using a gradient solvent system (*n*-hexane/ethylacetate) until the reaction was complete. The products were purified by recrystallization and silica gel column chromatography using gradient solvents (*n*-hexane/ethylacetate). The temperature of the reaction mixture was in the range of 120–180 °C. These compounds are not labeled for use and are still being investigated (Figure 7).

2,6-bis(morpholinomethyl)-4-nitrophenol (**LQM312**). M.p. 108–110 °C. Yield 85%. Reaction time: 10 min. IR (cm^−1^; CHCl_3_ film) 3376, 3026, 2952, 1725, 1600. ^1^H-NMR (200 MHz; DMSO-d6; Me_4_Si, δH): 8.02 (2H, s), 4.7 (1H, s, OH), 374–3.72 (8H, m), 3.70 (4H, s), 2.58–253 (8H, m). ^13^C-NMR (δC): 163.29, 138.57, 124.13, 123.18, 65.97, 57.23, 52.26. FAB-MS (M + 1) 338 (7%). Calculated for C_16_H_23_N_3_O_5_. C 51.73%, H 6.78%, N 11.31%, O 12.92%, S 17.26%.4-(*tert*-butyl)-2,6-bis(thiomorpholinomethyl)phenol (**LQM319**). IR (CHCl_3_ film) cm^−1^ 3456 (OH), 3197 (Csp2-H Ar), 2886 (Csp3-H). ^1^H NMR (CDCl_3_) d: 10.33 (1H, s, OH), 7.18 (1H, dd, J = 8.4 Hz, 2.7 Hz), 6.94 (1H, d, J = 2.7 Hz), 6.74 (1H, d, J = 8.4 Hz), 3.70 (2H, s, Ar-CH_2_), 2.82 (4H, m, -S-CH_2_-), 2.71 (4H, m, -N-CH_2_-),1.27 (9H, CH_3_). ^13^C NMR (CDCl_3_) d: 155 (C), 141.8 (C), 125.60 (CH), 125.49 (CH), 119.77 (C), 115.47 (CH), 62.51 (Ar-CH_2_), 54.36 (-N-CH_2_-), 33.84 (C), 31.48 (CH_3_), 27.79 (-S-CH_2_-). FAB-MS *m*/*z* (M + 1) 266 (80%), 265 (100%),163 (45%).4-nitro-2,6-bis(piperidin-1-ylmethyl)phenol (**LQM345**). Recrystallized from ethyl acetate yellow plates. mp 109–111 °C. 80% yield. IR (CHCl3 film) cm^−1^, 3450 (OH), 3150 (Csp2H Ar), 2903 (Csp3H). ^1^H NMR (CDCl_3_) δ ppm: 8.93 (1H, s, OH), 8.07 (2H, s), 3.81 (2H, s, Ar-CH2), 2.66 (4H, m), 1.72 (4H, m, CH_2_); 1.52 (2H, m, CH_2_). ^13^C NMR (CDCl_3_) δ ppm: 164.74 (C), 138.93 (C), 125.83 (CH), 121.44 (C), 57.91 (Ar-CH_2_), 53.34 (CH_2_), 24.84 (CH_2_), 23.37 (CH_2_). FAB-MS *m*/*z* (M + 1) 334 (35%), 240 (100%, M − C_5_H_10_N), 226 (23%).

### 4.2. High-Salt Diet

A total of 222 grams of starch, 10% (*w*/*w*) NaCl, and a finely ground regular rat diet [58] were mixed to make one kilogram of the final powder. Then, water was added until a homogeneous paste was formed. The paste was molded into pellets of a similar weight and shape to the original rat diet pellets, after which it was left to dry and harden.

### 4.3. Biological Models

This study used healthy male Wistar rats (*Rattus norvegicus*) (250 ± 25 g) housed in transparent polycarbonate boxes in groups of 6 animals per cage for each treatment in a controlled temperature room at 23 °C with an artificial 12 h light/12 h dark cycle for four weeks.

The animals were divided into control (control) and hypertensive (HP) groups, and in turn, each group was subdivided into the Captopril, LQM312, LQM319, and LQM345 subgroups. The control animals were maintained on a regular diet (0.4% NaCl) [58] and tap water ad libitum throughout the study. Moreover, HP animals consumed the high-salt diet (10% NaCl) previously described and tap water ad libitum throughout the study. The time of the experiments was 4 weeks, and the treatment with the compounds was administered through the fourth week intragastrically. Under our experimental conditions, no extra beats or other ventricular rhythm disturbances were noted.

### 4.4. Evaluation of the High Blood Pressure Model

During the four weeks of a high-salt diet, water, food intake, and weight were measured during the experimental time to ensure a high arterial blood pressure model. The amount of urine was obtained using metabolic cages, where the animals had been handled and habituated previously.

Arterial blood pressure was measured at the beginning of the experiment and two days before the euthanasia date. Animals were handled, prewarmed (32 °C) for 20 min, and subjected to restraint habituation to obtain reliable results. Then, they were introduced into an acrylic tube with a hole to access the animals’ tails. Blood pressure and heart frequency values were obtained using a tail-cuff method in conscious animals [59], transduced to a pneumatic pulse (Narco Bio-Systems Inc., Austin, TX, USA), and a programmed electrosphygmomanometer. After acclimatization, six consecutive measurements were performed on each rat with a one-minute interval between repetitions. The procedure had a precision of 7% (model 79, Grass Medical Instruments, Quincy, MA, USA). Data are reported in mmHg for blood pressure and beats per minute for the heart frequency.

### 4.5. Captopril and LQM Stock Solutions

Stock solutions (100 mg/mL) were prepared using 100 mg of each of the four compounds used and solubilized by adding 300 µL of hydrochloric acid solution (0.1 N) and 0.7 mL of injectable water to obtain a final volume of 1.0 mL. The therapeutic dose of 1 mg/kg/day in a sugary solution as the vehicle was administered intragastrically to each animal. The compounds were administered during the previous week before euthanasia. Captopril (N-[(S)-3-mercapto-2-methylpropionyl]-L-proline, cat. PHR1307) was purchased from Sigma–Aldrich. Pharmaceutical Secondary Standard; Certified Reference Material).

### 4.6. Isolation of Cardiac Myocytes

Rats were anesthetized with pentobarbital (63 mg/kg) and injected with heparin (0.05 mL), both i.p. Hearts were excised and washed with a warm (37 °C) isotonic sodium chloride solution (0.9%). Cardiac cells were obtained as previously described [36]. The Trypan blue method (Gibco BRL, Waltham, MA, USA) was used to determine cell viability and morphology [60].

### 4.7. Experimental Procedure

Aliquots of 4 × 10^4^ cells (100 mg) were centrifuged and resuspended independently in 100 µL of one of the following solutions:Basal conditions: Normal Tyrode.G1: Normal Tyrode with a goat polyclonal IgG anti-GLUT1 antibody directed to the extracellular domain (SC-1603, Santa Cruz Biotechnology, Santa Cruz, CA, USA) at a dilution of 1:1000 in Tyrode.G4: Normal Tyrode with a goat polyclonal IgG GLUT4 antibody directed to the extracellular domain (SC-1606, Santa Cruz Biotechnology, Santa Cruz, CA, USA) at a dilution of 1:10,000 in Tyrode.G4 for HP cells in LQM345 repetition experiments: Normal Tyrode with a goat polyclonal IgG GLUT4 antibody directed to the extracellular domain (SC-1606, Santa Cruz Biotechnology, Santa Cruz, CA, USA) at a dilution of 1:500 in Tyrode.

After 60 min of incubation (37 °C), a final soft mixing step ensured an equal glucose concentration, and the mixture was centrifuged again. The supernatants were stored at −20 °C to determine the glucose concentration further.

The experiments were carried out under oxygenated conditions with the solution aerated with 95%O_2_-5%CO_2_ (pO_2_ of 160 and pCO_2_ of 21.3 mmHg) and under hypoxic conditions with the solution aerated for 2 min with 95%N_2_-5%CO_2_ (pO_2_ of 21.3 and pCO_2_ of 27.3 mmHg).

The antibody concentrations were obtained by a dose–response curve from a previous study [37].

### 4.8. Determination of Glucose Uptake

The glucose concentration was determined by the glucose oxidase method [61]. A total of 2 µL of the supernatant and 200 µL of the Trinder reagent were added to a microplate and then incubated for 30 min. The results were read on an ELISA microplate reader at 505 nm (Benchmark Plus microplate spectrophotometer, Bio-Rad, Hercules, CA, USA). The absorbances were interpolated to a glucose concentration curve.

Cell glucose uptake was calculated by subtracting the glucose concentration in the supernatant after experimentation from the initial glucose concentration.

### 4.9. Molecular Docking and Molecular Dynamics Simulation

AMPK’s participation in various signaling pathways in the cardiovascular system and metabolic tissues, such as the adaptive response to hypoxia [62], allowed us to investigate this molecule as a candidate for GLUT4 trafficking [39].

The present study employed molecular docking techniques and molecular dynamics simulations to examine the interactions between the AMPK protein and the molecules LQM300s and Captopril.

The heterotrimeric structure of AMPK, composed of the alpha, beta, and gamma subunits in *Rattus norvegicus*, was constructed using sequences obtained from the UniProt platform [63]: the alpha subunit (Gen: Prkaa1, ID: P54645 AAPK1_RAT), beta subunit (Gen: Prkab1, ID: P80386 AAKB1_RAT), and gamma subunit (Gen: Prkag1, ID: P80385 AAKG1_RAT).

To predict the 3D folding of the entire protein, the advanced AlphaFold tool, specifically ColabFold v1.5.2-patch: AlphaFold2, was used [64]. The prediction process did not incorporate interactions or predefined templates; all other parameters remained at their default settings. A single seed was employed to generate the structure, selecting the structure with the highest quality and precision.

The molecules AMP, Captopril, LQM312, LQM319, and LQM345 were modeled using the Molecular Operating Environment 2022.02 (MOE) software [65]. This software was used for the preparation and execution of molecular docking data. The docking protocol was as follows: the initial pose search was performed using the triangle matcher algorithm, evaluated with the London dG scoring function, and the best 300 poses were retained. These poses were then optimized by allowing flexibility to the receptor and assessed with the generalized-born volume integral/weighted surface area scoring function (GBVI/WSA dG), which allows estimating the binding free energy of the ligand from a docking binding mode. The molecular docking simulation was performed in triplicate.

A representative docking pose was selected and prepared for molecular dynamics simulations using MOE for each compound. The protein–ligand complex was immersed in a cubic box with water under periodic boundary conditions (P1, 95 Å per side), and Na^+^ and Cl^−^ ions were introduced for charge neutralization. Subsequently, the structural minimization of the entire system was performed. Input files were generated with the Nanoscale Molecular Dynamics 2.13 software (NAMD) [66], employing the Amber 14 force field. The simulation protocol consisted of a gradual heating stage from 0° to 300° K over 0.5 ns, followed by an equilibrium stage at 300° K and 1 atm for 4 ns, culminating in a 100 ns production stage, where both pressure and temperature were kept at 300° K and 1 atm. The molecular dynamics simulations used a cutoff of 12 Å and a time step of 0.002 fs.

### 4.10. Statistics

Statistical analysis was performed using GraphPad Prism version 5.03 and SPSS version 21.0. Data are reported as means and standard errors (SEMs), n = 6.

Two-tailed Student’s *t*-tests were used for variables with normal distribution to compare differences between the control and HP groups, and *p* < 0.05 indicated statistical significance.

Differences among multiple groups were compared by a one-way analysis of variance (ANOVA), with a post hoc Bonferroni test to determine which differences were statistically significant at *p* < 0.05. The ANOVA for blood pressure had two factors: group (control or hypertensive) and treatment (Tyrode, captopril, LQM312, LQM319, or LQM345). The ANOVA for glucose uptake had four factors: group (control or hypertensive), treatment (Tyrode, captopril, LQM312, LQM319, or LQM345), hypoxia (oxygenated or hypoxic), and transporter (Basal, G1, or G4). The ANOVA for the glucose uptake of LQM345 with two different concentrations of G4 on hypertensive rats had two factors: hypoxia (oxygenated or hypoxic) and transporter concentration (Basal, G1, G4 1:1000, and G4 1:500). Statistical parameters, such as the root mean square deviation (RMSD) and root mean square fluctuation (RMSF), were calculated for each molecular dynamic simulation.

## 5. Conclusions

In summary, new antihypertensive medications are needed to improve the management of HP and its consequences. Our results show that Captopril and the LQM312, 319, and 345 compounds possess antihypertensive properties. However, only Captopril, LQM312, and 319 have cardioprotective properties. Captopril, an ACE inhibitor, reduces blood pressure and enhances glucose metabolism in the heart, providing cardioprotective benefits in hypertension. At the same time, LQM312 and LQM319 protected the heart during high blood pressure and low oxygen levels by modulating glucose transporters. The compounds are a good strategy for improving cardioprotection.

Computer-based investigations of molecular interactions yielded a comprehensive understanding of how these chemicals engage with the AMPK protein, a crucial therapeutic target in regulating blood pressure and cellular metabolism. Captopril and LQM312 were shown to have specific binding sites, indicating their strong attraction to the gamma subunit of AMPK and the ATP site located between the γ- and β-subunits. The results of this study indicate the possible mechanisms by which these chemicals may regulate AMPK activity and the glucose transporters’ participation in protecting the heart from hypoxia and hypertension. Although it was experimentally determined that the compound LQM319 produces a cardioprotective effect, no significant results were found during the simulation with AMPK to suggest LQM319 as an activator of AMPK via its binding to the AMP site. In contrast, LQM345 had promising characteristics as an AMPK inhibitor, as evidenced by its notable affinity for the ATP site and the lack of effect to promote the GLUT4 trafficking to the membrane, as presented in Figure 8. The findings of this study confirm the ongoing interaction between chemicals and AMPK in physiological circumstances, supporting the importance of these studies in developing new drugs for the treatment of cardiovascular disorders.

## Figures and Tables

**Figure 1 ijms-25-08812-f001:**
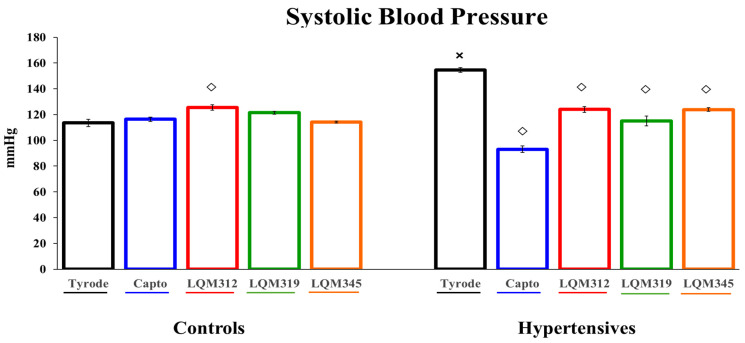
The antihypertensive effect of Captopril and LQM compounds in animals with and without HP. The values are expressed as mean ± SEM, n = 6. Two-way ANOVA (factors are group and treatment) with post hoc comparisons corrected by the Bonferroni method. **×**: *p* < 0.05 indicates statistical significance vs. the control group only in the untreated group. **◇**: *p* < 0.05, indicates statistical significance vs. the Tyrode group within controls and hypertensives separately.

**Figure 3 ijms-25-08812-f003:**
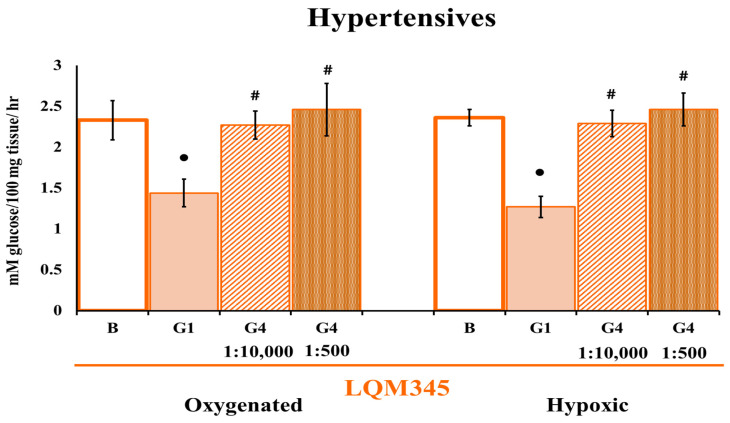
Cardiomyocyte glucose uptake by LQM345 pretreatment at a different GLUT4 antibody concentration. Effect of LQM345 on glucose uptake and GLUT4 participation in HP cardiac cells under oxygenation and hypoxia. The values are expressed as mean ± SEM, n = 6. Two-way ANOVA (factors are hypoxia and transporter) with post hoc comparisons corrected by the Bonferroni method. •: *p* < 0.05, statistically significant antibody effect vs. its corresponding basal uptake; **#**: *p* < 0.05, a statistically significant difference between GLUT 1 and GLUT 4 antibodies in both separate conditions. The ratios 1:10,000 and 1: 500 are GLUT4 antibody dilutions.

**Figure 4 ijms-25-08812-f004:**
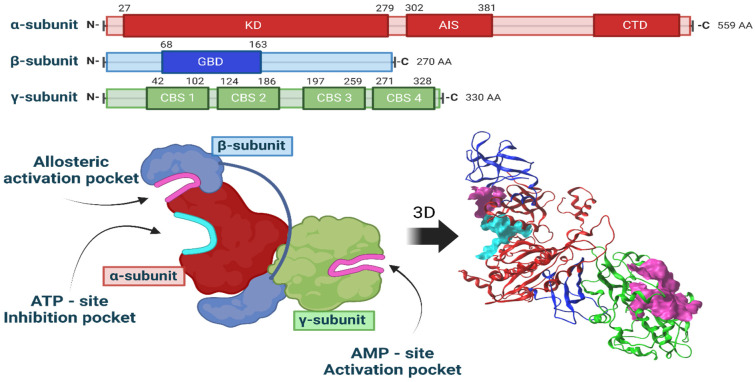
Three-dimensional model of the AMPK heterotrimer and representation of the sites of interest.

**Figure 5 ijms-25-08812-f005:**
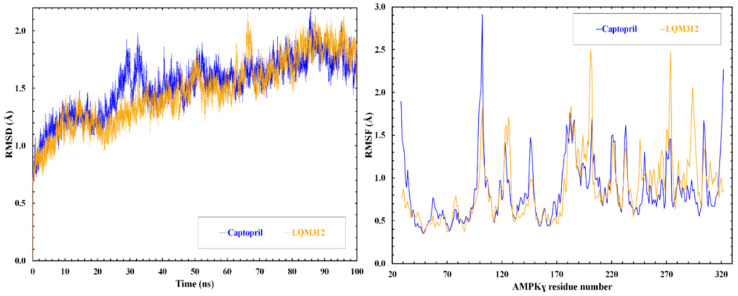
RMSD and RMSF of the γ-subunit during the molecular simulation.

**Figure 6 ijms-25-08812-f006:**
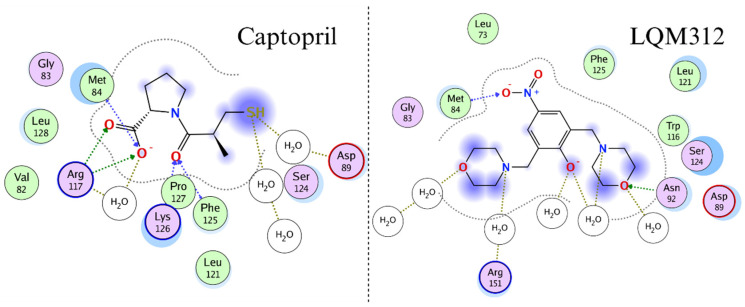
Two-dimensional representation of the interactions of the two compounds with the AMPK γ-subunit in molecular dynamics.

**Figure 7 ijms-25-08812-f007:**
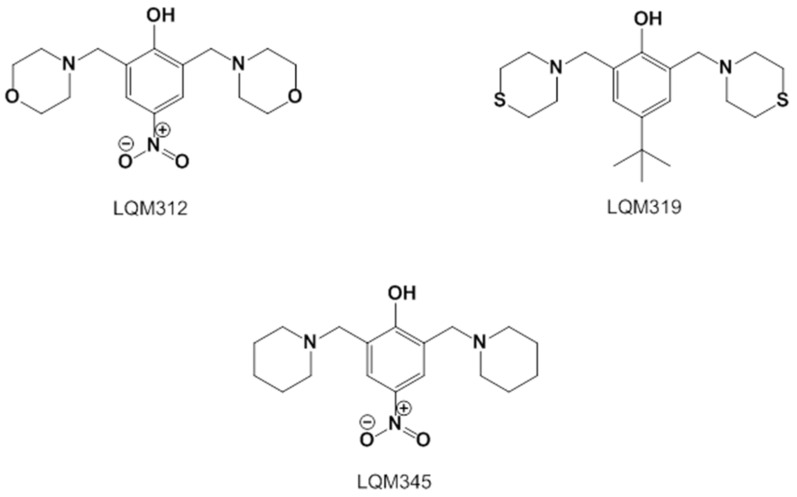
Structures of the experimental LQM compounds. **LQM312**: morpholine structure, **LQM319**: thiomorpholine structure, and **LQM345**: piperidinyl structure.

**Figure 8 ijms-25-08812-f008:**
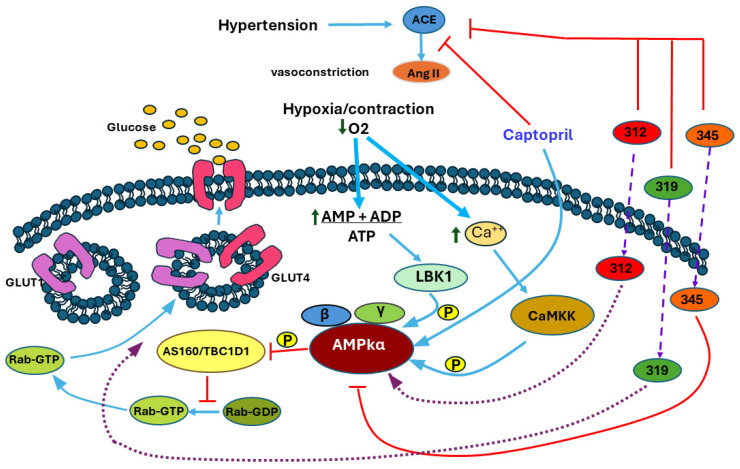
The mechanism proposed by the results obtained. The red lines represent inhibitions, the blue actions, the wide violet slashed lines indicate entrance through the membrane, and the purple dotted lines represent proposed mechanism based on in silico prediction.

**Table 1 ijms-25-08812-t001:** High-salt HP model and control characteristics. The values are expressed as mean ± SEM after one month of salt intake. Paired Student’s *t*-test, * *p* < 0.05, n = 24.

Parameters	Control	HP
Weight (g)	280 ± 4.6	303.0 ± 5.2 *
Systolic pressure (mmHg)	113.5 ± 2.8	152.02 ± 1.5 *
Diastolic pressure (mmHg)	85.6 ± 1.9	93.0 ± 3.2 *
Heart rate (beats/min)	394.3 ± 0.1	393.14 ± 0.4 *
Food (g)	22.5 ± 2.4	35.6 ± 5.2 *
Water (mL)	34.4 ± 4.4	168.5 ± 27.2 *
Urine (mL)	3.8 ± 0.5	19.2 ± 2.8 *

HP animals developed high arterial blood pressure and maintained it even if they stopped consuming a high-salt diet. The data in this table are from the fourth week when the HP was established.

**Table 2 ijms-25-08812-t002:** Docking scores for each compound at two AMPK sites.

Ligand	AMP Site Score (γ-Subunit) kcal mol^−1^	ATP Site Score (α-Subunit) kcal mol^−1^
AMP	−4.91 ± 0.59	−3.80 ± 0.41
Captopril	−3.26 ± 0.65	−2.68 ± 0.24
LQM312	−3.65 ± 0.27	−3.51 ± 2.7
LQM319	*	−4.06 ± 0.65
LQM345	*	−4.41 ± 0.54

* No statistically significant binding modes of LQM319 and LQM345 were identified at the AMP site.

**Table 3 ijms-25-08812-t003:** Ligand properties from Swiss ADME.

LIGAND		LIPINSKI RO5
**LQM 312**	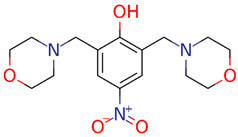	MW: 337.37 g/mol cLogP(XLOGP3): 0.32 HA: 7 HD: 1 RB: 5
**LQM 319**	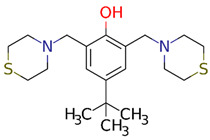	MW: 380.61 g/mol cLogP(XLOGP3): 3.76 HA: 3 HD: 1 RB: 5
**LQM 345**	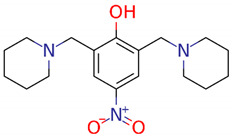	MW: 333.43 g/mol cLogP(XLOGP3): 2.76 HA: 5 HD: 1 RB: 5

MW, Molecular weight; cLogP, Calculated coefficient partition octanol/water; HA, Hydrogen bond acceptor; HD, Hydrogen bond donor; RB, Rotatable Bonds.

## Data Availability

The authors will make the raw data supporting this article’s conclusions available upon request.

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
