# Peer review of "Hypoxic Cardioprotection by New Antihypertensive Compounds in High Salt-Diet Hypertensive Rats: Glucose Transport Participation and Its Possible Pathway"

_ijms, 2024, doi:10.3390/ijms25168812_

Round 1
Reviewer 1 Report
Comments and Suggestions for Authors
The manuscript titled "Hypoxic cardioprotection by new antihypertensive compounds in high salt-diet hypertensive rats: Glucose transport participation and its possible pathway" presents an interesting study on the cardioprotective effects of novel compounds derived from Changrolin under hypoxic conditions in hypertensive rats. The research focuses on the role of glucose transporters GLUT1 and GLUT4 in this context. While the study provides valuable insights into the potential of these compounds for treating hypertension and related cardiac issues, there are areas that could be improved to strengthen the manuscript.
1. Clarity and Organization: The manuscript would benefit from a clearer structure and organization. The provided chunks are not always chronological or thematically grouped, which can make it challenging for readers to follow the narrative and understand the progression of the study. A more logical flow of information could improve readability and comprehension.
2. Methodological Detail: The manuscript should provide more detailed information on the experimental methods used, particularly regarding the synthesis of the LQM compounds and the in silico calculations. This would allow for better reproducibility and validation of the results by other researchers.
3. Statistical Analysis: The manuscript mentions the use of two-way ANOVA and Bonferroni tests for statistical analysis. However, it is not clear how the data were processed or if any other statistical methods were considered. A more thorough explanation of the statistical approach, including justification for the chosen methods and any corrections for multiple comparisons, would be beneficial.
4. Comparison with Existing Treatments: While the manuscript compares the novel compounds to captopril, a well-known antihypertensive, it would be helpful to include a broader comparison with other existing treatments. This would provide a more comprehensive understanding of the compounds' efficacy relative to the current standard of care.
5. Mechanism of Action: The manuscript suggests that the LQM compounds may mimic the interaction between AMP and the AMPK subunit, potentially activating the GLUT4 trafficking signaling pathway. This hypothesis requires further exploration and experimental validation to establish a clear mechanism of action for the compounds.
6. Side Effects and Safety: The manuscript does not discuss the potential side effects or safety profiles of the novel compounds. Information on any adverse effects observed in the study or predicted based on the compounds' structures would be important for assessing their clinical viability.
7. Clinical Relevance: The manuscript could benefit from a discussion on the clinical relevance of the findings. This includes potential applications of the compounds in human hypertension treatment, the feasibility of their synthesis at scale, and any regulatory considerations.
8. Conclusion: The manuscript should conclude with a strong summary of the findings, their implications for the field, and suggestions for future research directions. This would help to reinforce the significance of the work and guide subsequent studies.
Comments on the Quality of English LanguageMinor editing of English language required
Reviewer 2 Report
Comments and Suggestions for Authors
Comments to the authors
Hernández-Serda et al. investigated the antihypertensive effect of Captopril and three LQM compounds in hypertension rats kept on a high-salt diet. Using isolated cardiomyocytes from these animals, the authors measured glucose consumption under oxygenated and hypoxic conditions and determined whether GLUT1 and GLUT4 glucose transporters were involved in this process. In addition, the group performed computational analysis of the anti-hypertensive compounds binding the AMPK, modulating glucose uptake.
Although the work gives insight into interesting aspects on the effects of the anti-hypertensive compounds, there are points that need to be addressed to improve this manuscript.
Major points
The authors need to more specifically define the aim of the study and to provide rationales for the experiments carried out.
Minor points
It is not so clear when and how long were the animals treated with the Captopril and three LQM. The authors should add a short description to the results section how the treatment with the drugs was done.
Were the isolated cardiomyocytes also treated with the Captopril and three LQM compounds in the experiments measuring the glucose uptake.
The authors should show individual points in the bar graphs (Figure 1, Figure 2, Figure 3).
Reviewer 3 Report
Comments and Suggestions for Authors
Very interesting work that highlights the characteristics of antihypertensives/GLUTs. Points: 1- the protective action of Captopril is also known from the point of view of xardioprotection against ischemia/reperfusion in hypertensive models. The authors could take this article as reference: doi: 10.1007/s00395-009-0075-6. 2- the effects demonstrated by the authors are very interesting, from a therapeutic point of view glyphozines have been administered for many years now. The same glyphozines find the activation of AMPK on their way. Given the role of glyphozines, could the authors consider this aspect in the conclusion of their study? 3- how many animals were used? Was the size of the cells between the two groups changed following the treatments? 4- the expression of the total AMPK protein and its phosphorylation changed in the groups, it would therefore be appropriate to highlight this modification by western blotting of the different conditions 5- has the redox activity of the mitochondria changed? Has phosphorylation at the mitochondrial level changed?
6- to make the mechanism clearer it would be appropriate for the authors to include a graphical abstract
Reviewer 4 Report
Comments and Suggestions for Authors
I have read your timely paper with interest. In general, the material that you present may provide interesting new information regarding the actions of selected antihypertensive compounds on cardiac myocyte glucose transport in an adult rat model of dietary salt-induced hypertension. However, the main new findings (Fig. 2) in your manuscript are not put in a useful context and certainly are not discussed in sufficient detail or even in a format that is required for publication along with this series of papers. If you choose to revise, please consider:
1. Making no reference to the proarrhythmic aspects or possibilities of your findings. In your manuscript these concepts and references are dated and are likely misleading. The main conceptual updates that you are missing can be obtained from the two publications cited below and others.
Constitutively active adenosine monophosphate-activated protein kinase regulates voltage-gated sodium channels in ventricular myocytes.
Circulation. 2003 Apr 22;107(15):1962-5. doi: 10.1161/01.CIR.0000069269.60167.02. Epub 2003 Apr 7. PMID: 12682004
Cardiac mechanisms of the beneficial effects of SGLT2 inhibitors in heart failure: Evidence for potential off-target effects.
J Mol Cell Cardiol. 2022 Jun;167:17-31. doi: 10.1016/j.yjmcc.2022.03.005. Epub 2022 Mar 22. PMID: 35331696
2. The main weakness in your paper is the Discussion section. After reading and, in fact, rereading this section, I am at a loss to be sure of your main findings or their relationship to directly relevant previous work. I recommend a complete rewrite of this section along with additions of subtitles such as:
a) Summary of Major Findings
b) Relationship to Previous Results
c) New Mechanistic Insights
d) Possible Translational Applications
3. Your manuscript is well organized and the Figures are quite clear. However, in a number of key places you use very unconventional Scientific English and this leads to ambiguities. Examples include:
- line 45 'so'
- line 55 'a crucial protagonist'
- line 71 'high contraction'
- line 72 'better patient evolution'
- line 102 'to keep ATP levels'
In addition and as noted the entire Discussion needs to be edited, reorganized and rewritten.
Comments on the Quality of English LanguageMajor revisions required
Round 2
Reviewer 2 Report
Comments and Suggestions for Authors
We thank the authors for the response and revising the manuscript. The manuscript improved, however, there are still several points that need to be addressed.
Major points:
1) For better understanding, it would be important to add a short introductory sentence regarding the goal/rationale of experiments to paragraphs 2.1, 2.2, 2.3, 2.8 before proceeding with the description of the results.
2) Ll. 294-298: It is not so clear how were the results on the glucose transporters different from the hypothesis. What impact was expected for the compounds to have on the glucose transporters? This should be explained in more details to better understand why the authors modeled the interaction of compounds with possible mediators involved in the translocation of the glucose transporters to the membrane. Why did the authors focus on AMPK as the GLUT4 transporter and no other transporters? What about transporters associated with GLUT1 translocation?
3) If possible, it would be important to prove experimentally that Captopril/LQM compounds indeed interact with AMPK.
5) Based on the results of the study: which compound is the best to treat hypertension and improve glucose uptake?
Minor points:
1) 2.1 Paragraph: Please specify the HP characteristics in the text.
2) Table 1:
Were the parameters presented in table 1 determined 4 weeks after the high-salt diet or after another 4 weeks without high-salt diet? Please specify.
3) Figure 1 legend:
· L164: In the sentence: “X: P < 0.05 indicates statistical significance vs. the Control group”, what does control group include? All 5 conditions?
· L164: In the sentence: "rhombus: P < 0.05, indicates statistical significance vs. the Tyrode group”, what does tyrode group include? Both control and hypertension conditions?
4) For the paragraph 2.3 it would be important to add a small section in the beginning why the glucose uptake was measured and why oxygenated and hypoxic conditions were used.
Ll 170-171: In the sentence: “The participation of GLUTs in glucose uptake was determined by incubating cells with specific antibodies.” please specify “specific antibodies”. Do these antibodies block the function of the GLUT1/4?
5) Ll188-189: “During oxygenation, glucose uptake is mediated only by GLUT1, and under hypoxia, both transporters respond”. What does “both transporters respond” mean?
6) Figure 2 legend:
· l229: “Open circle”: P < 0.05, statistically significant vs. hypoxia. Which groups are meant, since the label is also present in “hypoxic” groups.
· L231: Rhombus: P < 0.05, a statistically significant effect of the compounds vs. Tyrode. Which tyrode is meant: in oxygenated or hypoxic tyrode conditions?
· L233: “black quadrat”: P < 0.05, statistically significant of the compounds vs. LQM319. Conditions are these: oxygenated or hypoxic, basal or G1 or G4?
· L231: X: P < 0.05, statistically significant controls vs. hypertensives. Which conditions are meant: oxygenate or hypoxic; tyrode, G1 or G4?
· L232: +: P < 0.05, statistically significant of the compounds vs. Captopril; “heart”: P < 0.05, statistically significant of the com-232 pounds vs. LQM312. Which conditions are meant: oxygenated or hypoxic; tyrode, G1 or G4?
7) ll.205: This sentence is not clear: “they could make the metabolic switch as Tyrode cells”.
8) Please add explanation regarding 1:10000 and 1:500 into figure 2 and 3 legend.
9) L. 281: The figure reference should be Figure 3?
Author Response
Thank you very much for all your suggestions, which enriched our work.

Reviewer 3 Report
Comments and Suggestions for Authors
The new version appears to be significantly improved. I agree
Author Response
Thank you very much for your enrichment. No file is attached.
Reviewer 4 Report
Comments and Suggestions for Authors
Thank you for considering each of my comments and responding meaningfully to most. My final suggestion is that your manuscript should very clearly point out that under your experimental conditions, no extra beats or other ventricular rhythm disturbances were noted. To me, this is a surprising but interesting finding.
Comments on the Quality of English LanguageThe writing and overall organization of the R1 manuscript have been improved significantly.
Author Response
The entire team would like to thank you for your suggestions, which have enriched this work and have been of great help to us.

Round 3
Reviewer 2 Report
Comments and Suggestions for Authors
We thank the authors for the response and revising the manuscript. The manuscript improved, however, there are still some points that need to be addressed.
Specifically, some explanations regarding the group comparisons were given in the author’s R1 response but not all added to the figure legends.
1) Figure 1 legend:
L. 167: rhombus: P < 0.05, indicates statistical significance vs. the Tyrode group.
This description is still not specified clearly in the legend that the comparisons are done for compounds vs. Tyrode within Controls and Hypertensives separately. Please add this to the figure legend. (This is explained in the author’s response but not in the figure legend).
2) Figure 2 legend:
a) Which groups are meant exactly in L244: “Open circle: P < 0.05, statistically significant vs. oxygenated? Are the corresponding groups compared?
E.g. Graph: Controls:
Does open circle shown for “Tyrode, B, hypoxic” indicate significance vs “Tyrode, B, oxygenated”?
Does open circle shown for “Tyrode, G4, hypoxic” indicates significance vs “Tyrode, G4, oxygenated”?
Does open circle shown for “LQM345, B, hypoxic” indicate significance vs “LQM345, B, oxygenated”?
Does open circle shown for “LQM345, G4, hypoxic” indicate significance vs “LQM345, G4, oxygenated”?
The sentence in L. 244 “Open circle: P < 0.05, statistically significant vs. oxygenated” needs to be better specified in the legend, otherwise it is not clear which groups were compared.
b) L. 246: Rhombus: P < 0.05, a statistically significant effect of the compounds vs. Tyrode.
It needs to specified in the figure legend that it is vs. its own Tyrode in both conditions. And which Tyrode condition is meant: B, G1, G4? Please specify.
c) L. 246: X: P < 0.05, statistically significant controls vs. hypertensives. Does it mean the corresponding groups of controls and hypertensives were compared? This also needs to specified in the figure legend.
d) Ll. 246-247: +: P < 0.05, statistically significant of the compounds vs. Captopril. “heart”: P < 0.05, statistically significant of the compounds vs. LQM312.
The information regarding “The significance symbols are positioned in the bar that is different from its corresponding group bars.” needs to be added to the figure legend. Are the comparisons done for the groups oxygenated and hypoxic conditions separately?
e) L248: “black quadrat”: P < 0.05, statistically significant of all GLUT4 groups vs. LQM319 GLUT4.
The information needs to be added to the figure legend: It is in both conditions, but only in the bar where it is positioned. That means that it is in the GLUT4 condition. Correction made on Line 248.
Are the oxygenated and hypoxic conditions compared separately?
Please add this information to the figure legend.
3) Figure 3 legend:
Ll. 315-316: •: P < 0.05, statistically significant antibody effect vs. its Basal uptake. #: P < 0.05, statistically significant difference between GLUT 315 1 and GLUT 4 antibodies. Are the oxygenated and hypoxic conditions compared separately? Please add this information to the figure legend.
4) Ll. 320.321: The authors state that “In our experimental conditions of the heart (hypoxia), glucose transport occurs through the insulin-independent AMP-activated protein kinase (AMPK) pathway.” This part of the sentence is not so clear: “our experimental conditions of the heart”, please rephrase it. Please add a reference for this sentence.
Author Response
Thanks for all your help. Our answers are in the file attached.
